# Gene Expression Studies in Down Syndrome: What Do They Tell Us about Disease Phenotypes?

**DOI:** 10.3390/ijms25052968

**Published:** 2024-03-04

**Authors:** Laura R. Chapman, Isabela V. P. Ramnarine, Dan Zemke, Arshad Majid, Simon M. Bell

**Affiliations:** 1Sheffield Children’s NHS Foundation Trust, Clarkson St, Sheffield S10 2TH, UK; laura.chapman38@nhs.net; 2Sheffield Institute of Translational Neuroscience, University of Sheffield, Glossop Road, Sheffield S10 2GF, UK; 3Sheffield Teaching Hospitals NHS Foundation Trust, Royal Hallamshire Hospital, Glossop Road, Sheffield S10 2GJ, UK

**Keywords:** Down syndrome, gene expression, brain, cardiac, haematopoietic

## Abstract

Down syndrome is a well-studied aneuploidy condition in humans, which is associated with various disease phenotypes including cardiovascular, neurological, haematological and immunological disease processes. This review paper aims to discuss the research conducted on gene expression studies during fetal development. A descriptive review was conducted, encompassing all papers published on the PubMed database between September 1960 and September 2022. We found that in amniotic fluid, certain genes such as *COL6A1* and *DSCR1* were found to be affected, resulting in phenotypical craniofacial changes. Additionally, other genes such as *GSTT1*, *CLIC6*, *ITGB2*, *C21orf67*, *C21orf86* and *RUNX1* were also identified to be affected in the amniotic fluid. In the placenta, dysregulation of genes like *MEST*, *SNF1LK* and *LOX* was observed, which in turn affected nervous system development. In the brain, dysregulation of genes *DYRK1A*, *DNMT3L*, *DNMT3B*, *TBX1*, *olig2* and *AQP4* has been shown to contribute to intellectual disability. In the cardiac tissues, dysregulated expression of genes *GART*, *ETS2* and *ERG* was found to cause abnormalities. Furthermore, dysregulation of *XIST*, *RUNX1*, *SON*, *ERG* and *STAT1* was observed, contributing to myeloproliferative disorders. Understanding the differential expression of genes provides insights into the genetic consequences of DS. A better understanding of these processes could potentially pave the way for the development of genetic and pharmacological therapies.

## 1. Introduction

Down syndrome (DS) is a well-known aneuploid condition caused by complete or partial trisomy of chromosome 21 (T21). The incidence is approximately 1 in 700 births and increases with higher maternal age [1]. It is associated with various clinical manifestations such as cognitive deficits, congenital heart defects, endocrine, gastrointestinal and immunological abnormalities, typical facies, sleep apnoea syndrome and an increased risk of certain diseases, including early onset Alzheimer’s disease and leukaemia [2]. Although it is established that the extra chromosome 21 is responsible for the DS phenotype, the specific genetic determinants of the individual clinical features are not fully understood. As such, DS is thought to be caused by multiple genes, with a generalised disruption of early developmental pathways [3]. It is unknown how many of the ~300 genes on chromosome 21 have any phenotypic effect when present in three copies [4]. The Down syndrome critical region (*DSCR*) has been extensively studied, but it is not solely responsible for the full DS phenotype [5]. Olson et al. have performed chromosome engineering of the orthologous mice segment to the human DSCR to confirm that non-contiguous genes may interact to produce the classical facial dysmorphology seen in the condition [5]. Therefore, studies that have focused solely on the DSCR may not fully understand the DS phenotype.

Research aimed at developing therapeutics for DS has been stalled due to the lack of clarity regarding the specific genes that cause the phenotypes associated with the condition. One particular phenotype of interest is that related to the amyloid precursor protein (*APP*). APP plays a crucial role in the pathogenesis of Alzheimer’s disease, as it serves as the precursor to the amyloid-beta protein that forms amyloid plaques, a pathological hallmark of the disease. Significantly elevated levels of *APP* expression have been observed in trisomy 21 trophoblasts. Additionally, partial duplication of the *APP* locus on chromosome 21 and rare cases of trisomy involving *APP* have been found to be causative factors in the development of early-onset Alzheimer’s disease neuropathology [6,7,8]. 

There are many viable ways of investigating genetic or whole-genome gene expression changes in DS mice or fetal cell models. These approaches include microarray analysis of DS cells with validation via real-time PCR, examination of the DNA methylation changes and RNA-sequencing technology. Each of these methods has its own strengths and weaknesses when it comes to identifying key gene targets. While numerous studies have detected a large number of differentially expressed genes, consistent changes in specific genes, sets of genes or pathways have not been identified to suggest a causative role. However, one issue with these studies is that the majority of them have been conducted in adults or children. It is important to note that the developmental changes leading to DS occur early in fetal development. Neural differentiation, for example, takes place from approximately 10 weeks and continues in the hippocampus and cerebellum after birth [9]. By understanding the gene expression changes at different stages of fetal development, researchers may identify potential targets and the optimal timing for future gene therapies.

This review aims to assess the studies conducted on gene expression in DS and identify candidate targets for gene therapy. Specifically, it will focus on studies using fetal tissue to investigate how changes in gene expression affect fetal growth, organ development and the expression of the DS phenotype. Additionally, studies utilising mouse models will be reviewed to understand how alterations in gene expression lead to differences in cellular function and structural changes. The review will begin by highlighting the different techniques used to analyse gene expression in DS samples and their respective strengths and weaknesses. The gene expression changes across different stages of fetal development will then be discussed, followed by an exploration of specific changes observed in different DS tissues and organs. Finally, the review will discuss how the information gathered from these studies can be applied to develop future gene therapies.

## 2. Methods

The studies included in this review were identified through searches conducted on databases including PubMed, Embase and Google Scholar, spanning from September 1960 to September 2023. The reference lists of the identified articles were also searched to find any additional relevant publications. A narrative overview of the literature was created by synthesising the findings from the literature retrieved through computerised database searches and manual searches. The free-text search terms used were ((Down syndrome) OR (DS) OR (Trisomy 21) OR (T21)) AND ((Microarray) OR (Gene Expression) AND (fetal)). The inclusion criteria included papers in the English language, or that could be translated, with the relevant search terms and relevant abstracts. Papers were excluded if full-text access was unavailable, if they did not report clear outcomes or if they could not be translated into the English language. 

## 3. Assessment of Gene Expression Analysis Techniques

Several methods have been employed to analyse gene expression differences in DS mouse models or fetal cells. Mouse chromosome 16 exhibits synteny with both human chromosomes 21 and 22 and, therefore, includes some genes from the 22q11 region. Initial modelling efforts utilised trisomy 16 mice, which displayed numerous defects shared with DS, such as cardiac septal defects. However, these models presented challenges as the triplicated genes were from the Mmu16 region and did not have complete homology to the trisomy of human chromosome 21 (Hsa21). Moreover, these animals often died at birth, making it challenging to investigate other dysmorphic features [10]. More recent DS mouse models include partial trisomy of 16, such as Ts1Cje, which carry additional genetic modifications that may impact the phenotypic presentation of DS differently from a human with DS [11,12]. Table 1 provides a summary of these different methods. The discrepancies in the study methods make it difficult to directly compare and validate the genes that have been identified using mouse models. 

Microarray analysis of DS fetal cells and tissues has been extensively investigated, but as a standalone tool, it may have limitations in accuracy. These limitations stem from challenges in interpreting copy number variations of unknown significance, incomplete penetrance or variable expressivity in the absence of a clear phenotype. Additionally, obtaining sufficient RNA for high-quality microarray results from prenatal samples can be challenging [13]. Real-time PCR is often used to validate the results from microarray analysis, improving accuracy. For instance, Shi et al. observed that certain genes identified through the Affymetrix assay were not validated by real-time PCR, underscoring the limitations of microarray analysis alone [14]. MicroRNAs play a crucial role in regulating embryonic development, cell differentiation, proliferation and apoptosis, making their expression assessment useful for understanding the genetic changes in DS [14]. Microarray analysis has become a more common method for assessing relevant microRNAs in DS. 

Single-cell RNA sequencing allows for the investigation of cellular transcriptomics and gene expression profiling [15]. This analysis is frequently employed to study the molecular pathophysiology underlying DS and other aneuploidies. In trisomic cells, single-cell RNA sequencing has revealed that the additional allele is transcribed independently. The “specific transcriptional profile for each gene contributes to the phenotypic variability of trisomies” [16].

DNA methylation undergoes a dynamic process involving both de novo methylation and demethylation during development. It plays a significant role in genomic imprinting, X-chromosome inactivation and transposition when DNA is dysregulated. DS has a profound impact on DNA methylation, particularly in haematopoietic cells early in life, making it one of the most studied forms of epigenetic regulation in DS [17]. Bisulphite sequencing is considered the “gold standard” in DNA methylation studies. It uses techniques such as methylation-specific PCR, PCR and sequencing and bead array. Bead arrays are a cost-effective approach that allows for the identification of specific regions of interest [17]. Manufacturers claim that these assays can detect DNA methylation levels as low as 0.5% using PCR, making them highly accurate for the quantification and identification of tissue-specific biomarkers [18]. DNA methylation is believed to explain some of the multiple phenotypes observed in DS [19]. One common hypothesis suggests that the increased expression of specific genes on chromosome 21 accounts for increased methylation of these phenotypes through gene dosage effects [19].

Quantitative transcriptome map analysis integrates gene expression profiles from different tissues, providing an overview of changes in entire organs. Validation of gene expression changes can be achieved through RT-PCR, which enhances reliability and allows for an overview of multiple organs and tissues. Antonaros et al. utilised the transcriptome mapper (TRAM) software [20] to create a T21 blood cell transcriptome map and employed the Samtools software to read and identify maps on the HR-DSCR [21]. TRAM software has been utilised to compare transcript expression levels and profiles between DS and normal brain, lymphoblastoid cell lines, blood cells, fibroblasts, thymus and induced pluripotent stem cells. Each gene analysis technique has its own relative advantages and disadvantages. Utilising multiple methods can help validate gene expression changes in DS fetal tissues and improve the accuracy of identification.

**Table 1 ijms-25-02968-t001:** A summary of the advantages and disadvantages of different techniques used for analysis.

Technique	Advantages	Disadvantages
Microarray analysis	Results can be validated using real-time PCR Allows for expression levels of thousands of genes at once	Poor accuracy due to difficulty interpreting copy number variants of unknown significance [13] Limited to genomic sequences Problems with probe cross-hybridisation or sub-standard hybridisation
DNA methylationanalysis	Highly sensitive—can detect DNA methylation levels as low as 0.5% Very accurate in quantification Increases understanding of gene regulation and identifies potential biomarkers [22]	Multiple different methods of analysing DNA methylation with some disadvantages for each method
Quantitative transcriptome map	Allows an overview of changes in a whole organ Can be further validated by RT-PCR	Inappropriate for identifying genes with large impacts on adaptive responses to the environment [23] mRNA abundance is an unreliable indicator of protein activity [23] Standard practice in the analysis is limited by prioritising highly differentially expressed genes over those that have moderate fold changes and cannot be annotated [23]
Western blot	High sensitivity, able to detect 0.1 ng of protein, which can be used in early diagnosis [24] High specificity due to gel electrophoresis and the specificity of the antibody–antigen interaction [24]	Time-consuming, non-quantitative process Skilled analysis and laboratory equipment required as a minor error in the process can cause incorrect results and false negatives if proteins are not given enough incubation time [24] Primary antibodies needed can be expensive False-positive results due to antibodies reacting with a non-intended protein [24]
Immunohistochemistry	Relatively low cost [25] Quick Can be done on fresh/frozen tissue samples [25] Allows in situ verification of various antibodies at the same time in organs, tissues and cells [25]	Not standardised worldwide [25] The process is cheap, but the initial equipment to run it is expensive [25] It is non-quantitative [25] High chance of human error and relies on antibody staining optimisation [25]
Real-time PCR (RT-qPCR)	Measures RNA concentrations over a large range [26] Sensitive Processes multiple samples simultaneously [26] Provides immediate information [26]	Requires optimisation of good primers and correct reaction conditions [26]
Flow cytometry analysis	Fast single-cell multiparametric analysis Very accurate and can be used on very small populations of cells [27] Good at highlighting non-uniformity [27] Produces very detailed data [27]	Very slow analysis [27] More expensive than alternate assays [27] It is non-quantitative; it provides average densities but not specific amounts [27] Relies on antibody staining optimisation and requires very specialised instrumentation for the analysis
Single-cell RNA sequencing	Assesses quantification and sequence of RNA using next-generation sequencing (NGS) [28] Uses short reads of mRNA and reveals which genes are turned on [28] Allows detection of novel transcripts and is quantifiable [28]	Requires a large quantity of starting material to isolate sufficient high-quality RNA [28] Generates a large quantity of data which require complex analysis [28] RNA degrades rapidly Subjected to amplification bias [28]

## 4. Gene Expression Changes in Amniocytes and Amniotic Fluid

The screening process for DS includes a non-invasive prenatal test (NIPT), which isolates cell-free fetal DNA in the maternal blood. It aims to determine the likelihood of aneuploidy by assessing the aberrant copy number for whole chromosomes or segments of chromosomes specific to the test [29]. An invasive test such as amniocentesis, in which amniotic fluid (AF) is acquired, is still necessary to confirm the diagnosis for those identified to be at high risk of developing aneuploidy following initial NIPT screening [30]. AF can be split into two fractions: supernatant (cell-free components, placenta-derived microparticles, protein, cell-free fetal DNA and cell-free fetal RNA from the fetus) and amniocytes [29]. Amniocytes are cells that can be derived from several fetal tissues. These cells can be cultured and subsequently used for a variety of purposes. 

Chung et al. screened cultured amniocytes for expression changes using a custom array containing 102 genes on chromosome 21. Only the GSTT1 gene was differentially over-expressed [31]. GSTT1 is thought to play a role in carcinogenesis. A previous study by Chen et al. stated that the inheritance of the GSTT1 null genotype conferred a 4.3-fold increased risk of developing myelodysplastic syndromes [32,33]. In the amniocytes in DS cases, 2 genes out of 24 were down-regulated, COL6A1 and PRSS7. COL6A1 from the collagen superfamily plays a role in the integrity of tissues [34,35]. COL6A1 has been shown in previous studies to be downregulated in the brain but expressed in the atrioventricular (AV) canal. This change in expression is thought to contribute to AV-node-related cardiac defects [36,37,38]. COL6A1 has also been associated with Bethlem myopathy and therefore could be linked to the hypotonia and joint laxity of DS [39,40]. These studies demonstrate the potential for the AF transcriptome to reflect fetal and placental development. It could therefore assist in the monitoring of normal development [41].

Altug-Teber et al. cultured amniocytes and chorionic villus cells, focusing on chromosome 21. They found 33 and 16 overexpressed genes, respectively, with none showing under-expression [34]. Cultured amniocytes and chorionic villus sampling (CVS) provide a high yield of high-quality mRNA for the array. However, results must be interpreted cautiously because the mRNA is derived in an environment very different from that of the womb. One of the overexpressed genes was DSCR1. The DSCR, a segment on chromosome 21, contains genes responsible for various features of DS, including craniofacial dysmorphology [5]. A study by Saber et al., focused on DSCR4, highlighted that overexpression of DSCR4 in the neural crest cells, which account for over 90% of craniofacial development, specifically alters facial morphology in DS [42]. The DSCR1 gene on chromosome 21 is a developmental regulator and plays a role in neurogenesis. Its overexpression may contribute to brain abnormalities seen in DS [43]. 

Amniotic fluid supernatant has been used to detect genome-wide expression changes using Affymetrix microarrays [43]. A total of 414 probes showed significant changes in expression, with 5 of them present on chromosome 21 genes. The upregulated genes included CLIC6, ITGB2 and 2 ORFs (C21orf67 and C21orf86), whilst RUNX1 was downregulated. CLIC6 is a member of the chloride intracellular channel family of proteins and is involved in the activation of the cAMP-dependent PKA pathway, which regulates pathogenicity, hyphal growth and stress tolerance [44,45]. ITGB2 encodes an integrin beta chain that plays an important role in the immune response. If upregulated, it may contribute to the differences in immune response in people with DS. However, these results were not confirmed using an additional assay, revealing the limitations of the Affymetrix microarray probes. This was also demonstrated in the study by Rozovski et al., which highlights different expression profiles once validated [6]. Nonetheless, the samples were matched for sex and gestational age, improving accuracy, as these factors have previously been shown to impact results [45].

Huang et al. analysed metabolites present in individuals with DS using amniotic fluid [46]. The AF was processed, and metabolomic fingerprinting was conducted using ultra-performance liquid chromatography and tandem mass spectrometry (UPLC-MS). Alterations in porphyrin metabolism, bile acid metabolism, hormone metabolism and amino acid metabolism were validated for the two experimental sets. Significant changes were observed in the metabolites of coproporphyrin III, glycocholic acid, taurochenodeoxycholate, taurocholate, hydrocortisone, pregnenolone sulphate, L-histidine, L-arginine, L-glutamate and L-glutamine. Analysis of these metabolic alterations was linked to aberrant expression of chromosome 21 genes PDE9A, GART and FTCD. Specifically, the decrease in coproporphyrin III in the DS fetus may be linked to abnormal erythropoiesis, and the unbalanced glutamine–glutamate levels were found to be closely associated with abnormal brain development in the DS fetus. It is important to note that the study had a small sample size of 10–15 controls and cases, which reduces the statistical power.

Studies of AF cannot only diagnose DS but also highlight the specific organ systems or tissues that might be affected after birth. As there is a highly heterogeneous expression of the DS phenotype in individuals with the condition, further research could explore whether the magnitude of the changes in gene expression associated with neural development, immune competency, or collagen stability correlates with the severity of the observed phenotype. This would potentially allow the development of individualised patient therapies if the risk of AF sampling could be justified. 

## 5. Gene Expression Changes in the Placenta

The placenta, specifically the chorionic villus, is commonly used for prenatal testing. Placental tissue can also be obtained in the event of pregnancy termination following a diagnosis of DS. 

Gross et al. used seven 2nd trimester placentas from fetuses with T21 and seven matched and seven non-matched cDNA samples from normal karyotype placentas as controls [47]. They used microarray technology to evaluate differences in gene expression. Their custom array contained approximately 9000 cDNA clones. About 643 cDNAs were found to be overexpressed in T21 compared to controls and 3 cDNAs were found to be under-expressed. When compared with age-matched controls, only 13 differentially expressed cDNAs were found. The use of microarrays for prenatal placental samples has been slow due to potential difficulties interpreting copy number variations of unknown significance and incomplete penetrance or variable expressivity in the absence of a clear phenotype [13]. This study highlights that different genes were expressed in the age-matched controls compared to the non-age-matched controls, demonstrating the dynamic nature of gene expression during gestation and the importance of studying different time points and using age-matched controls.

A more recent study by Lee et al. focused on finding novel epigenetic markers on chromosome 21 that exhibit a hypermethylated pattern in fetal placenta compared to blood using PCR [13]. They performed a high-resolution tiling array analysis of chromosome 21 using a methylated-CpG binding domain protein-based method. They identified 93 epigenetic regions that showed placenta-specific differential methylation patterns. Three regions showed fetal placenta-specific methylation patterns in T21 placenta samples. These three regions were detectable with high diagnostic accuracy as early as the first trimester, as confirmed by further statistical analysis. Therefore, these studies demonstrate clear genetic changes in the placenta and help increase our understanding of aetiology, potentially aiding in the identification of targets for treatment.

Rozovski et al. measured the detectable expression of 5334 genes out of over 10,000 on an oligonucleotide microarray, using cultured trophoblasts derived from placental samples obtained through prenatal testing in the first trimester [6]. The sample consisted of four normal male fetuses and four T21 males. They found that 65 genes were significantly altered in the DS cases, with 51 over-expressed and 14 under-expressed, after correction for multiple comparisons. The three genes with the highest significant fold change were MEST, SNF1LK and LOX. MEST is believed to play a role in development and is usually expressed in mesoderm derivatives [48]. Cultured trophoblasts were used for ethical reasons and to ensure sufficient and high-quality DNA, as it can be challenging to obtain from prenatal samples [13]. The results of this study were validated using qRT-PCR, which improved the accuracy of the findings. This highlights some differences, such as MAT2A being under-expressed in microarray and over-expressed in qRT-PCR. This is due to the non-specific hybridisation of MAT2A transcripts to Affymetrix microarray. The probe on the Affymetrix array shares complete identity with a sequence in the early endosome antigen 1 gene on chromosome 12, leading to skewed expression and highlighting a limitation of the Affymetrix microarray. Despite this limitation, the study identifies specific genes that are over-expressed in DS compared to controls. These findings can be compared with other studies to increase their power and further studied to determine their viability for diagnosis or as therapeutic targets.

Studies of gene changes in the placenta highlight the dynamic nature of gene expression during gestation and underscore the importance of studying different time points in development. While there are ethical issues with studying early tissue, however, the gene changes identified in these studies are invaluable in ongoing research for potential biomarkers. 

## 6. Gene Expression Changes Affecting Brain Development

The development of the brain is drastically affected in DS and is the cause of the most prominent hallmarks of the disease. There are known changes in gross brain structure and microdysgenetic changes. These manifest as hypotonia at birth, abnormal gait, ligamentous laxity, seizures, intellectual disability and early neuropathological changes in Alzheimer’s disease (often by 40 years old) [49].

El Hajj et al. investigated changes in gene expression in the developing DS fetal cortex, using the frontal cortex of 16 DS and 27 controls and the temporal cortex from 8 DS and 8 controls [50]. They used Illumina 450K arrays which showed that 1.85% of all analysed CpG sites were significantly hypermethylated and 0.31% hypomethylated in the fetal DS cortex. Methylation values were quantified using Pyro Q-CpG software ((PyroMark Assay Design 2.0 software (Qiagen, Manchester, UK)) and methylation PCR. Specifically, there was reduced expression of NRSF/REST due to upregulation of DYRK1A (21qq22.13). REST is a transcriptional repressor involved in the repression of neural genes. DYRK1A is a dual-specific tyrosine phosphorylation-regulated kinase that has been implicated in various processes critical to neurodevelopment [51]. It acts synergistically to dysregulate NFATc transcription factors, which are regulators of vertebrate development [51,52]. These changes are linked to intellectual disability, speech development and autism, potentially explaining some of the phenotypical features of DS [51,52,53]. Methylation of REST binding sites during early development may contribute to a genome-wide excess of hypermethylated sites. There was upregulation of DNMT3L, which is hypothesised to lead to de novo methylation of neuroprogenitors, persisting in the fetal DS brain, while DNMT3A and DNMT3B are downregulated in DS. DNMT3A and DNMT3B are responsible for establishing DNA methylation patterns during embryogenesis, so this alteration will contribute to phenotypic characteristics in DS [22]. Another finding was that a large number of differentially methylated promoters are present on chromosomes other than 21. The PCDHG cluster on chromosome 5 is involved in the neural circuit formation in the developing brain. It was hypermethylated and downregulated in DS, resulting in a reduction in dendrite arborisation and growth in cortical neurons. This study used gestational age-matched controls, to reduce errors in result interpretation caused by differentiation in expression at different stages of development.

Shi et al. extracted total RNA from fetal hippocampal tissues to analyse miRNA and mRNA expression using Affymetrix miRNA 4.0 and PrimeView Human Gene Expression Array, which were validated by real-time PCR [14]. They found a specific repertoire of miRNAs involved in the hippocampus in trisomy 21. These included hsa-miR-138, hsa-miR-409 and hsa-miR-138-5, suggesting that their altered activity in the hippocampus was a causal factor for intellectual disability in DS. Further studies of miRNAs have been conducted by Deng et al. and Lim et al. [54,55]. Deng et al. found that miR-125b-2 on chromosome 21 is overexpressed in DS patients with cognitive impairment [54]. miR-125b-2 is known to promote specific types of neuronal differentiation; however, its full function in the developing embryo is unknown. The study looked at the overexpression of miR-125b-2 and found that it inhibited the differentiation of mouse embryonic stem cells (mESCs) into endoderm and ectoderm and impaired all-trans-retinoic acid-induced neuron development in embryoid bodies. Therefore, it is important to determine the rate at which the brain and neurons develop. Lim et al. examined the possibility of using miRNAs as potential non-invasive biomarkers for the detection of fetal trisomy 21 [55]. They used microarray-based genome-wide expression profiling to compare the expression levels of miRNAs in whole blood samples from non-pregnant women, pregnant women with euploid T21 fetuses and placenta samples from euploid or T21 fetuses. They found that 150 RNAs were up-regulated in the placenta in T21 and 149 were down-regulated. miRNAs mir-1973 and mir-3196 were expressed at higher levels in the T21 placenta compared to the euploid placenta. These miRNA studies are useful in increasing our understanding of the changes in a developing DS brain and how these changes cause the well-known phenotype.

The DSCR genes have been found to be differentially expressed in other studies. Esposito et al. found gene expression changes in neural progenitors derived from the frontal cerebral cortex. Expression of DSCR genes was increased in DS cases [56]. The study analysed 608 probes differentially expressed, representing 334 genes and 46 functional networks. Further analysis found that the upregulation of S100B and APP in this critical region activates the stress response kinase pathways and is linked to the upregulation of aquaporin 4 (AQP4). Changes in AQP4 have been linked to epilepsy, oedema, Alzheimer’s disease and other CNS disorders, which may explain the phenotypical changes in DS [57,58,59]. It is important to highlight that not all patients with DS will have genetic changes in the DSCR region. A study by Lyle et al. used genomic hybridisation to analyse 30 patients with anomalies in chromosome 21. They found that five patients had trisomy 21 and this did not include the DSCR, thus highlighting that there are further gene changes beyond this region [60].

Olmos-Serrano et al. conducted a multi-region transcriptome analysis of DS and euploid control brains, looking from mid-fetal development (14 weeks post-conception) until adulthood (42 years old) [61]. This study was designed to explore the complexity of brain development and changes over many decades, describing how genes involved in brain development may modulate over time. Dysregulated genes are found throughout the genome and not solely on chromosome 21. The transcriptome profiling performed using total RNA was extracted from 11 regions, including multiple regions of the cerebral neocortex, hippocampus and cerebellar cortex, revealing a genome-wide alteration in the expression of a large number of genes. These genes exhibited temporal and spatial specificity and were associated with distinct co-expression networks, with distinct biological categories, providing novel insights into multiple biological processes affected in the developing and adult DS brain. The M43 gene aids in the regulation of action potential and axon ensheathment. It was found to be downregulated in the DS neocortex and hippocampus during development and associated with myelination. Myelination is one of the most prolonged neurodevelopmental processes, continuing until the third decade of life [62,63]. If this process is impacted, it implies that the neurodevelopmental process in DS continues throughout the first few decades of life. The analysis protocol used was standardised with high-quality post-mortem human brains, thus increasing the reliability of the results. Co-dysregulation of genes associated with oligodendrocyte differentiation and myelination was validated by cross-species comparison in Ts65Dn trisomy mice. 

A study conducted by Shimizu et al. discovered that the DS mouse model exhibited defective early neuron production during prenatal life and hippocampal hypoplasia [64]. To understand this better, transcriptomic analysis was performed on Ts1Cje mice, comparing their transcriptomic profile to that of prenatal forebrain at embryonic day 14.5. The analysis revealed a decrease in the expression of TBX1 mRNA in both the prenatal forebrain and adult hippocampus Ts1Cje mice. These findings were further validated in other DS mouse models, namely, the Dp (16)1Yey/+ (longer trisomic regions) and Ts1Rhr (shorter trisomic regions) mice. It is worth noting that the TBX1 region is present in individuals with DiGeorge syndrome, a condition associated with conotruncal outflow tract cardiac defects [65]. Hence, it can be hypothesised that TBX1 plays a role in cardiac defects in DS patients. However, despite both conditions presenting with cardiac defects and a decreased TBX1 expression, they are distinct from each other. DS differs from DiGeorge, as DS individuals present with endocardiac cushion AVSD and AV canal defects, whereas DiGeorge is more likely to present with interrupted aortic arch or truncus arteriosus. TBX1 is believed to be involved in delayed fetal brain development and postnatal psychiatric phenotypes in DS. Another noteworthy finding is the dysregulation of the interferon-related molecular networks in the hippocampus of Ts1Cje mice, leading to the overexpression of Ifnar1 and Ifnar2 genes [64]. This dysregulation may contribute to the immunodeficiency observed in DS.

Olig2 is a crucial basic helix-loop-helix transcription factor involved in mammalian CNS development and is located in the critical region of trisomy 21 [66]. In a previous study by Shimizu et al., it was observed that Ts1Cje mice had increased Olig1 and Olig2 gene expression compared to wild-type mice [64]. However, the specific phenotypic features resulting from these changes remain unclear. Liu et al. focused on this specifically by developing an Olig2-overexpressing transgenic mouse line with a Cre/loxP system. This led to the development of microcephaly, cortical dyslamination, hippocampus malformation and profound motor deficits [66]. The authors also detected extensive neuronal cell death and downregulation of neuronal specification factors (Ngn1, Ngn2 and Pax6) in the developing cortex of mice misexpressing Olig2. Additionally, Olig2 was found to be significantly upregulated in the frontal cortices of individuals with DS at gestational ages of 14 weeks and 18 weeks [66]. Chromatin-immunoprecipitation and sequencing analysis confirmed that Olig2 directly targets the promoter and/or enhancer regions of Nfatc4, Dscr1/Rcan1 and Dyrk1a, the critical neurogenic genes that contribute to Down syndrome phenotypes and inhibit their expression. Another study by Chakrabarti et al. specifically examined Olig1 and Olig2, two genes triplicated in DS [67]. They created a DS mouse model, Ts65Dn Olig1/2+/−, by breeding Ts65Dn mice with Olig1/2+/− mice to restore the disomic expression of Olig1 and Olig2 genes. It was observed that inhibiting Olig1 and Olig2 rescued the inhibitory neuron phenotype in the Ts65Dn brain, underscoring the importance of these genes in cognitive development. 

Another gene implicated in oligodendrocyte proliferation is SLC35A2 [68]. It has recently been found to have an impact on focal cortical dysplasia, which involves extensive oligodendrogliosis in the subcortical white matter [68]. This gene could potentially be upregulated in DS, making it a promising target for future studies.

Comparatively little is known about gene changes causing epilepsy in DS. A study conducted by Takashima et al. used Golgi stains to highlight a progressive retraction of terminal dendrites in the molecular zone of the cerebral cortex, loss of dendritic spines and defective cortical layering in 14 newborn and older infants with DS [69]. The low incidence of epilepsy in DS, except from infantile spasms, may be explained by these changes. The abnormal development of neurons in the visual cortex of human fetuses and infant dendritic synapses is characterised by glutamatergic and excitatory features, resulting in a reduction in excitatory input without a corresponding alteration in axo-somatic inhibitory synapses. This leads to a shift in the excitatory/inhibitory ratio that favours inhibition [70]. The synaptic ratio of excitatory/inhibitory afferents is crucial in explaining epileptogenesis at a neuronal level and contributes to the lack of seizures in the immature brain development in DS [70]. Another study by Sarnat et al. focused on the importance of the proteoglycan (keratan) barrier in the developing human forebrain [71]. This barrier isolates cortical epileptic networks from deep heterotopia, insulates axonal fascicles, and explains why axosomatic synapses are inhibitory. However, the genetic cause of this has not yet been proven. Pathological spine loss, as observed in DS, can be explained by changes in the expression of signalling proteins, such as Rho-GEF (KALRN), Rho-GAP (OPHN1), Cdc42-GEF (ARHGEF9) and Rac-GAP (OCRL1) [72,73]. Investigating these proteins, specifically within DS, in future studies could shed light on genes that contribute to the loss of dendritic spines. It is believed that this loss of spines may also impact intellectual disability [74]. Torres et al. focused on the importance of TSP-1, which is secreted by astrocytes [74]. Astrocytes form part of the blood–brain barrier and aid in neuronal pathfinding, metabolic processes and synaptic transmission. TSPs are thought to play a role in synaptogenesis. A post-translational deficiency in TSP-1 was found in DS, leading to an alteration in dendritic spine structure and a reduction in spinal and synapse numbers [75]. This finding holds significant implications for future studies investigating the causes of developmental delay in DS.

These studies increase our understanding of the phenotypical manifestations of DS, including intellectual ability and the neuropathological changes in Alzheimer’s disease. The identification of differentially expressed genes may be attributed to the various techniques used and the different tissues analysed. Thus, the purpose of this review is to analyse the advantages and limitations of the different techniques. It is important to note that many of the studies are of small scale, due to ethical reasons such as obtaining fetal tissue. These considerations must be taken into account when interpreting results.

## 7. Gene Expression Changes Affecting Cardiac Tissues

Congenital heart defects (CHDs) occur in approximately 50% of individuals with DS, with the most common being atrioventricular septal defect (AVSD) [76]. The exact mechanism behind their occurrence is currently unknown. In a study conducted by Li et al., approximately 10,000 genes were screened in both heart tissue and skin fibroblast cultures [77]. In the DS case, 110 genes were differentially expressed in the heart, with 17 of them located on chromosome 21. In skin fibroblast cultures, 7 genes on chromosome 21 were differentially expressed, all of which showed increased expression. Among these genes, GART displayed the highest overexpression in fibroblasts. GART encodes a trifunctional enzyme that catalyses the de novo inosine monophosphate biosynthetic pathway, which is involved in de novo purine synthesis. Normally, GART is highly expressed during prenatal development of the cerebellum but becomes undetectable shortly after birth. However, in individuals with DS, the expression of GART and related proteins remains high postnatally [78]. Previous studies have also observed elevated serum purine levels in individuals with DS [78,79]. This increase in purine levels may contribute to phenotypical features such as intellectual disability, hypotonia and increased sensorineural deafness observed in DS [78,79].

MiRNAs are believed to have an important role in regulating cardiac development. Five miRNAs located on chromosome 21, miR-99a-5p, miR-125b-2-5p, let-7c-5p, miR-155-5p and miR-802-5p, have been previously studied. However, their expression in trisomy tissues has not been explored. Izzo et al. conducted a study that revealed the downregulation of miR-99a-5p, miR-155-5p and let-7c-5p in the hearts of DS fetuses with trisomy. Additionally, it was found that let-7c-5p and miR-155-5p are involved in mitochondrial function [80]. Since mitochondrial dysfunction is characteristic of DS, these miRNAs might impact mitochondrial dysfunction and cardiogenesis [81].

In a study by Bosman et al., a human embryonic stem cell model of DS was employed [82]. The results of their study suggested the involvement of two candidate genes on chromosome 21, ETS2 and ERG, in the disruption of the secondary heart field development and the consequent occurrence of CHD AVSDs, as they were found to be overexpressed during the early stages of cardiogenesis. These genes could serve as potential targets for gene therapy in the future. The use of the sibling hESC model allowed for the replication of early cardiogenesis in DS, considering variations in differentiation and disease development itself rather than slight discrepancies in genetic or epigenetic backgrounds. Furthermore, an electrophysiological abnormality was observed in the function of T21 cardiomyocytes, which correlated with mRNA expression data obtained through RNA-Seq. This finding warrants further investigation to determine its reproducibility.

A study by Liu et al. examined mouse mutants carrying various genomic rearrangements in syntenic regions of human chromosome 21 (Hsa21) [83]. They discovered that a triplication of the Tiam1-Kcnj6 region on mouse chromosome 16 (Mmu16) led to cardiovascular abnormalities associated with DS. To further investigate DS-related heart defects, Liu et al. generated two tandem duplications encompassing this region, using recombinase-mediated genome engineering. They found that Dp (16)4Yey duplication, spanning 3.7 Mb from Ifnar1 to Kcnj6, resulted in heart defects, whereas Dp(16)3Yey triplication, covering 2.1 Mb from Tiam1 to Il10rb, did not. Consequently, they identified the 3.7 Mb genomic region as the smallest critical genomic region related to DS-associated heart defects. 

The aforementioned studies employ different methods and utilise different DS models. Interestingly, cardiac neural crest cells normally contribute to the development of the cardiac septum and the Purkinje conduction system in infants. The gene expression changes observed in the mentioned studies do not involve genes expressed in neural crest cells. Thus, it is imperative to conduct further research specifically focusing on gene expression in neural crest cells in DS, as these may cause the septal defects seen in DS [84]. To determine whether targeting these gene expression changes singularly or as a group is necessary to prevent the cardiac alterations observed in DS, larger-scale studies would need to be conducted.

## 8. Gene Expression Changes That Lead to Haematopoietic Cells/Myeloproliferative Disease

Several studies have demonstrated that individuals with DS have an elevated risk of developing certain cancers. Children with DS have a 10- to 20-fold higher relative risk of developing acute leukaemia compared to the general population [81,85,86]. This strongly suggests a link between DS and the neoplastic formation of haematopoietic cells, specifically in the megakaryocyte lineage cells. A recent extensive cohort study revealed that children with DS were 2.8% more likely to be diagnosed with leukaemia, in contrast to only 0.05% of children without DS [87].

Chiang et al. conducted a study to determine if silencing trisomy through XIST gene induction could largely normalise haematopoietic phenotypes associated with DS [4]. XIST is an X-linked gene responsible for the natural inaction of X chromosomes in human female cells [88]. In their study, they inserted a doxycycline-inducible full-length XIST cDNA into one of three chromosomes 21s in induced pluripotent stem cells (iPSCs) derived from a male DS individual. They discovered that XIST induction in four independent transgenic clones consistently corrected the over-production of megakaryocytes and erythrocytes, which are implicated in myeloproliferative disorder and leukaemia. 

Further research identified specific genes that were over-expressed in DS and are associated with myeloproliferative disease. Kubota et al. conducted an integrated genetic/epigenetic analysis and found hypermethylation of RUNX1 on chromosome 21 in DS-acute lymphoblastic leukaemia (ALL), but not in ALL without DS [89]. Muskens et al. conducted an epigenome-wide association study on neonatal blood spots and discovered that the top two differentially methylated regions overlapped with RUNX1 and FLI1 [90]. Both studies employed DNA methylation analysis carried out using a chip-based analysis. The utilisation of blood spots allowed for a large sample size compared to other studies, incorporating 196 DS patients and 439 controls. Analysis of the cohort was also adjusted for heterogeneity, increasing the study’s power. RUNX1 plays a crucial role in blood cell differentiation, particularly in B cells [91]. Therefore, hypermethylation of RUNX1 may be associated with a higher incidence of B-cell precursor ALL in DS patients. 

A study conducted by Belmonte et al. examined the gene SON on chromosome 21 [92]. They observed reduced expression in a zebrafish homologue of SON, which resulted in decreased production of red blood cells, fewer brain and spinal malformations, reduced thrombocytes and myeloid cells and a significant decrease in T cells. This finding may provide insights into the immunodeficiency and myeloproliferative disorders that arise in DS. In DS, immune system abnormalities include T- and B-cell lymphopenia, reduced specific antibody responses to vaccines, defects of neutrophil chemotaxis, a decrease of naïve lymphocytes and impaired mitogen-induced T-cell proliferation [93]. Ishihara et al. conducted a transcriptomic and flow cytometry analysis of the E14.5 Ts1Cje mouse embryo brain to study the impact of multiple genes on the myeloproliferative system [94]. They observed increased neutrophil and monocyte ratios in CD45-positive haematopoietic cells, as well as a decrease in macrophages. This analysis involved multiple methods, including microarray, informatics analysis, validated with quantitative RT-PCR, Western blotting, flow cytometry, immunohistochemistry and image analyses, along with in vivo BrdU labelling. The DNA microarray analysis of the E14.5 Ts1Cje embryo brain revealed elevated expression of S100a8, S100a9, MPO and Ly6c1 mRNAs, which are abundant in neutrophils and/or monocytes. They also discovered that the triplication of Erg plays a role in the self-renewal of haematopoietic stem cells and haematopoiesis in the liver during embryogenesis [95,96]. Erg triplication in DS contributes to the dysregulation of the homeostatic proportion of immune cell populations in the embryonic brain and decreased prenatal cortical neurogenesis. It should be noted that this study only focused on male mice; therefore, these changes may not be consistent in female mice with DS. Previous studies have indicated the presence of sex-specific abnormalities, suggesting a possibility of sex-specific phenotypic features of DS. Therefore, it would be necessary to include female mice in future research [45]. 

Other studies have also focused on the mechanism in DS that causes immunodeficiency. A recent study by Kong et al. investigated 45 DS patients and observed elevated levels of IFN-αR1, IFN-αR2 and IFN-γR2 expression on the surface of monocytes and EBV-transformed B cells [97]. Furthermore, they found consistently high levels of total and phosphorylated STAT1 (STAT1 and pSTAT1) levels in unstimulated as well as IFN-α- and IFN-γ-stimulated monocytes from DS patients. However, these levels were lower compared to individuals with GOF STAT1 mutations, as GOF STAT1 mutations result in enhanced cellular response to IFN. DS participants exhibited normal levels of Th17, a high proportion of terminally differentiated CD8+ T cells and low levels of STAT1 expression. These findings contribute to our understanding of the mechanisms underlying immunodeficiency in DS patients [98]. Table 2 provides an overview of key genes in different organ systems that could be targeted for gene therapy in DS.

The increased incidence of certain cancers, such as acute leukaemia, in DS individuals highlights a significant genetic difference. These studies also enhanced our understanding of how these gene alterations can impact the immune system in DS. 

## 9. Gene Expression Changes That Lead to Endocrine Disease

There are several common metabolic changes in DS patients, including an increased risk of obesity, diabetes mellitus and dyslipidaemia [107]. However, many studies have been primarily focused on genes encoded by chromosome 21 that may influence these metabolic changes, rather than directly studying them within a DS cohort.

Parra et al. conducted a study specifically investigating RCAN1, a gene located on chromosome 21 [103]. RCAN1 inhibits calcineurin, a calcium-activated protein phosphatase important in cardiac remodelling. By downregulating RCAN1 in both neonatal and adult cardiomyocytes, the mitochondrial network becomes more fragmented. To compare the mitochondrial network between DS and disomic controls, induced pluripotent stem cells derived from DS patients were used. The study revealed that DS mitochondria were more fused, and their oxygen consumption was higher [103]. This highlights the impact of genetic differences in DS, potentially contributing to the observed metabolic changes seen in DS. 

In another study by Panagaki et al., it was found that fibroblasts from DS individuals showed increased expression of CBS compared to control cells [104]. CBS is an enzyme involved in various metabolic pathways, including homocysteine, folate and transsulfuration. DS cells also exhibited suppressed mitochondrial electron transport, oxygen consumption and ATP generation. However, these abnormalities were normalised by pharmacological inhibition of this using siRNA-mediating silencing of CBS. Therefore, the upregulation of CBS indicates a profound metabolic change in DS individuals and may contribute to the complex metabolic changes observed in DS. CBS is also crucial for the production of hydrogen sulphide, a gaseous transmitter that, when overproduced, inhibits complex IV, leading to metabolic and neurological deficits associated with DS by suppressing mitochondrial oxygen consumption and ATP generation [108]. 

Chromosome 21 also encodes PFK, which catalyses the phosphorylation of fructose-6-phosphate to fructose-1, 6-biphosphate, a key regulatory hormone in glycolysis [105,106]. Transgenic mice overexpressing PFK were found to have increased glucose utilisation in the brain, similar to the faster glucose metabolism observed in young DS brains, which is hypothesised to be linked to cognitive disabilities [105]. Additionally, PFK was found to be elevated specifically in embryonic PFK liver-type mice, suggesting that this altered glucose metabolism may contribute to changes in the early dysregulation of brain development [109].

Further research is needed to investigate specific gene changes that impact the metabolic alterations observed in DS. Chromosome 21 also encodes other proteins, such as BACE2, RCAN1 and DYRK1A, known to be associated with diabetes mellitus phenotypes and potentially contribute to the increased risk of diabetes mellitus [110,111,112].

## 10. Gene Expression Changes Affecting Ocular Development

There are various ocular manifestations associated with DS, such as strabismus, amblyopia, nystagmus, accommodation deficits, nasolacrimal duct obstruction, keratoconus, optic nerve pathology, neoplastic disease and retinal pathology [113]. A review highlighted that keratoconus, a corneal dystrophy leading to progressive visual impairment, was reported by 0–71% of individuals with DS [113]. It is important to note that the cornea and brain have the highest concentration of keratan sulphate proteoglycan in the human body [114].

In their study, Akoto et al. examined various ophthalmic manifestations of DS, the genetic factors relating to the cornea, central corneal thickness and mechanical forces on the cornea. These factors play critical roles as risk factors in the pathophysiology of keratoconus and their association with DS [115]. Furthermore, the analysis revealed a genetic association between keratoconus and DS through sequence variants within or near the COL6A1 and COL6A2 genes on chromosome 21. In addition to this, it has been suggested that mutations in SOD1, which is located on chromosome 21 and directly involved in antioxidant defence, may also link DS with this ocular manifestation. This is due to the significant accumulation of oxidative stress observed in the corneas of patients with keratoconus [116].

## 11. Gene Therapy for Future Implications

If an individual gene or group of genes is conclusively determined to play a role in DS, the question arises of what forms of intervention could be used. Prevention, although a more logical course of action, would require treatment early in fetal development. While individual genes can be inhibited through methods such as small interfering RNAs, the situation becomes more complicated when multiple genes, large chromosomal regions or entire chromosomes are involved.

Furthermore, when considering that DS can result from partial trisomies of chromosome 21 and that patients can have mosaicism of the chromosome, the practical challenges of using this type of genetic therapy become more complex. Additionally, ethical debates surrounding this topic are significant. In 2014, Inglis et al. conducted a study where they produced a questionnaire for parents to express their opinions on gene therapy for their child with DS [117]. Of the 101 parents involved in the study, 41% responded that they would treat their child, if possible, with 27% stating they would not. However, this study is limited in power due to its small sample size. Therefore, not only does gene therapy need further development, but it also needs to be approached in an ethically responsible manner. Modulating disease risk rather than stopping DS development in utero may be a more accepted gene therapy strategy for individuals living with DS.

## 12. Conclusions

In this review, we have highlighted how studying gene expression at the earliest stages of DS development can identify genes worth further investigation, as they may play causative roles in the DS phenotypes. We have also emphasised how the genes could be targeted in the future to modulate or prevent disease in individuals living with DS, opening up potential therapeutic avenues for this common genetic cause of intellectual disability. However, several barriers still exist when studying gene expression changes in DS. As mentioned in various sections of this review, the expression of potential causal genes can change throughout development, even into early adulthood. Furthermore, studies can be limited by small sample sizes due to ethical constraints, and ethical considerations surrounding DS treatment must be considered. Genetic therapies could potentially fundamentally change what it means to be a person living with DS. Future work would benefit from trying to understand what causes the heterogeneity in the expression of the condition, and how to silence key genes to prevent the individual disease features in DS. As gene therapy progresses, it will also be crucial to involve individuals with DS in the discussion. The complex ethical, medical and scientific questions that will arise from this line of research will have a significant impact on individuals with DS and their families.

## Figures and Tables

**Table 2 ijms-25-02968-t002:** A table summarising the different organ systems and detailing the gene expression changes seen in each tissue and also the different organ systems and the microRNA (miRNA) changes seen in each tissue. DSCR: Down syndrome critical region of chromosome 21; RT-PCR: reverse transcriptase PCR; qRT-PCR: quantitative reverse transcriptase PCR.

	Gene	Chromosome Position	Gene Expression Change	How This Affects Development
Nervous system	NRSF/REST [61]	4q12	Downregulated	Transcriptional repressor, represses neuronal genes in non-neuronal tissues [99]
	Ngn1 [66]	14	Downregulated	Neuronal cell death [66]
	Ngn2 [66]	4	Downregulated	
	Pax6 [66]	11	Downregulated	
	DNMT3A [61]	2q23	Downregulated	DNA methylation in the late stage of embryonic development [61]
	DNMT3B [61]	20q11.2	Downregulated	DNA methylation in a broader range of genes in early embryonic development [22]
	PCDHG [61]	5q31	Downregulated	Reduction in dendrite arborisation and growth in cortical neurons [61]
	M43 [62]		Downregulated	Regulation of action potential and axon ensheathment, neocortex and hippocampus over development [62]
	TBX1 [66]	HSA22q11	Downregulated	Fetal brain development and postnatal psychiatric phenotypes in DS [66]
	Olig1 [66]	DSCR	Upregulated	Microcephaly, cortical dyslamination, hippocampus malformation, profound motor deficits. Promotes enhancer regions of Nfact4, Dscr1/Rcan1 and Dyrk1a > DS phenotype [66]
	Olig2 [66]	DSCR	Upregulated	
	S100B [57]	DSCR	Upregulated	Activate the stress response kinase pathways and upregulated aquaporin 4 [57]
	APP [57]	DSCR	Upregulated	
	DYRK1A [50,61]	21qq22.13	Upregulated	Reduces NRSF/REST [50,61]
	DNMT3L [61]	21q22.4	Upregulated	De novo methylation in neuroprogenitors that persist in fetal DS brain [61]
	TSP-1 [74,75]	15q14	Downregulated	Alter dendritic spine structure, reduce spinal and synaptic numbers—causing developmental delay [74,75]
	GART [77]	21	Upregulated	De novo purine synthesis > intellectual disability, hypotonia, increased sensorineural deafness [78]
	ETS2 [83]	21q22	Upregulated	Most likely cause 2nd heart field development, AVSDs [83]
	Mmu16 [80]	16	Triplication	AVSDs [80]
Blood	SON [94]	21	Downregulated	Lower RBCs produced, brain and spinal malformations, reduced thrombocytes and myeloid cells, significant decrease in T cells [94]
	STAT1 [100]	2q32.2	Downregulated	Low = reduced enhanced cellular response to IFN [98]
	XIST [4]	Xq	Upregulated	X-chromosome inactivation in females Induction corrected the over-production of megakaryocytes and erythrocytes [88]
	RUNX1 [90,92]	21	Hypermethylation	Differentiation of blood cells, B cells Support bone cell development and differentiation [91]
	S100A8 [95]	1q21	Upregulated	Abundant in neutrophils/monocytes [95]
	S100A9 [95]	1q21	Upregulated	
	MPO [95]	17q12-24	Upregulated	Creates reactive oxidant species, part of the innate immune response, and contributes to tissue damage during inflammation
	Ly6c1 [95]	15	Upregulated	Part of the inflammatory response in atherosclerosis, regulates endothelial adhesion of CD8 T cells [101]
	IFN-αR1 [66,100]	21	Upregulated	Expressed on the surface of monocytes, EBV-transformed B cells and important in immunodeficiency [66,100]
	IFN-αR2 [66,100]	21	Upregulated	
	IFN-γR2 [100]	12	Upregulated	
	ERG [95]	21	Triplication	Self-renewal of haematopoietic stem cells and haematopoiesis in the liver during embryogenesis Dysregulation of the homeostatic proportion of the population of immune cells in the embryonic brain and decreased prenatal cortical neurogenesis [95]
	SOX2 [102]	3q26.33	Downregulated	Reduction in airway smooth muscle discontinuous in the proximal airway [102]
Lung	DYRK1A [102]	DSCR	Upregulated	Reduced incidence of solid tumours (neuroblastoma) and defects in angiogenesis of central arteries developing in the hindbrain [102]
Endocrine	RCAN1 [103]	DSCR	Downregulated	Important in cardiac remodelling and mitochondrial function [103]
	CBS [104]	21q	Upregulated	Enzymes involved in homocysteine, folate and transsulfuration pathways, mitochondrial electron transport and ATP generation [104]
	PFK [105,106]	12q13, 21q22 and 10p [106]	Upregulated	A key regulatory hormone in glycolysis [105,106]
Other	DSCR4 [42]	DSCR	Upregulated	Regulation of human leukocyte migration Craniofacial abnormalities [42]
	miRNA	Chromosome position	Gene expression change	How this affects development
Neuro	Hsa-miR-138 [14]	16q13	Upregulated	Hippocampus development [14]
	hsa-miR-409 [14]	14	Upregulated	
	hsa-miR-138-5p [14]	3 and 13	Upregulated	Intellectual disability [14]
	miR-125b-2 [55]	21	Upregulated	Cognitive impairment, promotes neuronal differentiation [55]
	mir-1973 [64]	21	Upregulated	Regulating CNS and nervous systems [64]
	mir-3196 [64]	20	Upregulated	
Cardiac	miR-99a-5p [81]	21q21.1	Downregulated	Congenital heart defects [81]
	miR-155-5p [81]	21	Downregulated	Mitochondrial dysfunction [81]
	Let-7c-5p [81]	21q21.1	Downregulated	

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
