# Peer review of "Gene Expression Studies in Down Syndrome: What Do They Tell Us about Disease Phenotypes?"

_ijms, 2024, doi:10.3390/ijms25052968_

Round 1
Reviewer 1 Report (New Reviewer)
Comments and Suggestions for Authors
In this review by Chapman et al., the authors discuss gene expression studies in Down Syndrome(DS) during fetal development. The review is unique in its subject matter. There are no other review articles covering specifically differential expression of genes in DS. They go over in detail gene expression in amniocytes, amniotic fluid, placenta, and different organ systems such as the nervous system, endocrine, etc.
It is well-written and very interesting. They cover all the relevant parts of the conducted research and highlight the advantages and limitations of each.
Methods are adequately described. The tables are a great way to summarize and compare different techniques/ expression patterns.
References are comprehensive and relevant.
It is important to research gene expression during development because of the impact that early disruption of these pathways has on phenotype and the different spatio-temporal expression patterns the same gene can have. This review provides a great summary of these important studies in one place.
Minor concerns:
There seem to be a few oversights regarding language, such as the use of "an" in line 42, "develop" in line 61, etc.
Authors use both "fetal" and "foetal" in the text. Usage should be uniform throughout the article. As far as I know, The spelling "fetus" is the preferred spelling in the medical world, regardless of location.
Comments on the Quality of English LanguageThere seem to be a few oversights regarding language, such as the use of "an" in line 42, "develop" in line 61, etc.
Author Response
We thank the reviewer for their kind comments regarding the review. Thank you for reviewing it and picking up a few minor things that needed changing. I have changed the language in line 42 and 61. I have also checked over the whole paper again, to change any other minor grammatical errors. As recommended I have also changed all foetal to fetal. Please see the updated manuscript.
Thank you again.
Reviewer 2 Report (New Reviewer)
Comments and Suggestions for Authors
Overall, the manuscript by Chapman et al., "Gene expression studies in Down Syndrome; what do they tell 3 us about disease phenotypes?" is fine. There are however, issues with both tables, they need to be reformatted. Table 1 is hard to read as is. Table 2 is spread over more than one page making it hard to understand, also the grey and non-grey bars do not make sense, I would suggest separating miRNA and genes.
Author Response
Thank you for taking the time to review our paper. We have edited the tables, so that they are on one page and some of the bullet points edited to make more sense. They grey and non-grey aspect has been removed, as we are aware this could be misleading. I have changed the landscape of the tables and separated miRNA and genes as suggested, hopefully it is now clearer. Please see attached manuscript to allow for reference of tables within the text.

This manuscript is a resubmission of an earlier submission. The following is a list of the peer review reports and author responses from that submission.
Round 1
Reviewer 1 Report
Comments and Suggestions for Authors
This is a valuable review of gene expression changes in Down syndrome, focusing on prenatal cells where the developmental changes of the extra 21 chromosome take place. Two additions would increase the value of the paper, the first reviewing briefly the historical focus and large grant funds expended on a critical region of chromosome 21 as the cause of Down syndrome. The authors clearly show the role of many genes outside of chromosome 21 in generating the phenotype and demonstrate the misguided focus of these prior studies. Second would be to include the disease links of some of the genes by including OMIM references from www.omim.org. COL6, for example, has been associated with Bethlem myopathy and would be of interest regarding the hypotonia and joint laxity of Down syndrome (Demon Genes May Deform Common Syndromes: Collagen VI Gene Change in Down Syndrome Unifies the Medical and Molecular Approach to Hypermobility Disorders. J Biosciences Med. 2020; 10:1-7. DOI: 10.4236/jbm.2022.103001).
The problem with including disease links is that this article really addresses research approaches to evaluating gene expression changes, so perhaps the clinical links could be explored in a follow-up article. However, brief attention to the prior fallacies of one chromosome region causing Down syndrome and its attendant mouse tri16 models is mandatory as it highlights the importance of their article.
Reviewer 2 Report
Comments and Suggestions for Authors
In this manuscript, the authors review the literature on gene expression studies in fetal development in Down syndrome. The authors discuss approaches used by different investigators and their results as they relate to a number of organ systems. There are many statements that are not referenced about the functions of specific genes identified in the reviewed studies. Differential expression of a gene does not prove function and statements about function need to be qualified by studies that provide more direct evidence of these statements. The authors are commended for trying to imagine therapeutic approaches to using this data, but the approaches proposed are based on cell culture studies of genetically targeted cell lines and really not approaches that will lead to therapies. This review tackles an important topic, but this manuscript needs a great deal of editing before consideration for publication.
If you are going to talk about APP, you should briefly mention the literature on partial duplications proving it is necessary (e.g., PMID 16369530) and rare cases of trisomy of APP alone prove sufficiency in causing early onset AD neuropathology (e.g., PMID 16959815). The argument is much stronger than overexpression in fibroblasts.
Line 88- The phrase “due to studies varying in power and validity” should be a new sentence. Do you mean some of the studies are not valid? Why review invalid studies? Were studies not sufficiently powered? Wouldn’t you then exclude them?
Line 157- How sure can anyone be that gene expression in amniocytes has anything to do with fetal development or “physical characteristics seen in DS?” I think it is fine to describe these studies, but to state that they provide any insight into fetal development without serious caveats is a disservice to the reader.
Line 182- The idea that results were not validated using another technique does not confirm the limitations of the assay- you may want to put in the caveat that that the study results were not confirmed using an additional assay.
Line 271- why does Dyrk1a trisomy lead to decreased expression of NRSF/REST? This was assumed to be true, but should be explained
Line 272- “Dyrk1a is a kinase which functions to help the development of the nervous system” is too vague to be useful. You can state that the kinase has been implicated in a variety of processes critical to neurodevelopment and then describe some or at least direct the reader to relevant reviews.
Lines 339-340- I wouldn’t assume that decreased expression of TBX1 causes cardiac abnormalities in DS. Either site a study that proves it, or raise a hypothesis, but it cannot be assumed.
The Olig2 paper presented to demonstrate the importance of Olig2 in neurodevelopment is one in which transgenic overexpression produces a phenotype more dramatic than the phenotype of Down syndrome and is therefore difficult to interpret. A better study to cite would be Tarik Hadar’s paper looking at rescue of Olig gene trisomy and restoration of aspects of neural development and physiology (PMID: 20639873).
Lines 372-373 This sentence is confusing. You can say that the identification of different differentially expressed genes may be due at least in part to the different techniques used and different tissues analyzed.
Line 375- why is it unethical to perform large scale studies? Do the authors intend to say that obtaining tissue is difficult? Do ethics issues arise in obtaining fetal tissue?
Line 386-387- Why does disrupting this pathway lead to these specific phenotypes? This sentence is vague.
Line 390-391- Why does overexpression of EST2 and ERG cause cardiac phenotypes in DS? The authors assume this based on the gene expression, but without more specific proof, I don’t think this statement should be made.
Line 402-404- Why is the discussion of synteny between mouse chromosome 16 and human chromosome 21 brought up so late in the review, after discussions of other studies using the mice have already occurred? A description of the fact that all of the mouse models studied share this in common should be established earlier in the text with the initial mention of the mouse models.
Line 452- This study is interesting, but unrelated to the topic of differential gene expression studies performed in foetal tissue. Was it a differentially expressed gene in one of the cited studies?
Line 473- What is the immunodeficiency phenotype in DS? Readers may not be aware of it.
Table 2 needs at least one reference for every stated function. Do the references associated with the genes listed indicate the gene expression studies in which they were found to be differentially expressed AND studies in which the consequence of the differential expression is proven? If not, the proposed functions must be based on data from the literature and cited.
I don’t understand why XIST is brought up as a potential therapeutic approach. In the stem cell model in which XIST is employed to inactivate a third chromosome 21, the cells were engineered to have XIST on one of the copies of chromosome 21. Are the authors suggesting that this should be attempted in patients? Has anyone tried to inactivate chromosome 21 using a technique analogous to X-inactivation? I wouldn’t propose it otherwise.
The authors reference a paper in which TK-NEO drug selection was used to induce disomy in cells that harbored the cassette on one chromosome, but it is not clear how this would be implemented as a therapeutic option. It would involve an extensive amount of cell culture, at which point genotyping blastocysts and only implanting disomic embryos seems to be a much more reasonable proposition.
Does XIST overexpression lead to overproduction of megakaryocytes? Is there real, cited data to support the claims presented in Figure 1? Based on the rest of this review, it is not clear.
Line 595- I don’t think that you can make the claim that these gene expression studies prove that any of the expressed genes do anything. They definitely identify genes that are worth studying further with the potential for playing causative roles in DS phenotypes, but I think this has to be clearly and carefully stated.
In addition, there are typographical errors that need to be addressed, including the following:
Line 22- “including” should be include or the sentence needs to be restructured
Line 26- were found to be dysregulated
Line 27- were dysregulated
Line 31- Down syndrome
Line 52- decide if you want to use fetal or foetal throughout the manuscript
Line 66- affect instead of will affect
Line 87- differences not discrepancies
Line 127- software not softward
Line 128- identify not indentify
Line 129- Transcriptome mapper (TRAM) software has been used
Line 130- comparison of
Line 145- amniocentesis,
Line 168- One gene cannot include one gene. Maybe state One overexpressed gene was… or something like that
Line 179- “this usually regulates” has to be reworded
Line 181- plays, not play. Also, either “the” immune response or immune responses
Line 194- aberrant expression of Chromosome 21 gene PDE9A…
Line 215- evaluate differences (Why feasibility? This is confusing)
Line 217- were found
Line 221- in the absence
Line 227- show, not shows
Line 228- using a
Line 233-235- this sentence is not clear- do the changes in the placenta hold promise for developing new diagnostic assays? How do they help with respect to aetiology?
Line 265-270. Maybe just use the word foetal once and then it is understood that all of the tissue are from foetal brains
Line 270- there was reduced…
Line 282- …a large number of differentially methylated promoters are present on…
Line 296- a large number of genes
Line 384- encodes
Line 408- They found, not stated. (The only thing that matters is the data)
Line 429-431- This sentence is difficult to read. State that trisomy 21 is seen in leukemias arising in disomic patients, suggesting a role in tumorigenesis, or something like that. Please include the incidence with which this occurs.
Line 471- there are
Comments on the Quality of English LanguageThere are a large number of typographical errors and grammatical errors. I tried to highlight many of them and supply suggested corrections.
Reviewer 3 Report
Comments and Suggestions for Authors
See attached Word file. I could not successfully copy and paste my comments here.

Reviewer 4 Report
Comments and Suggestions for Authors
in the doc

Round 2
Reviewer 2 Report
Comments and Suggestions for Authors
I appreciate that the authors have made substantial changes to the manuscript; however, I think there are too many errors remaining to consider publication at this time. I’ve listed a few here:
Line 55- Is APP a genetic change? It is a gene implicated in a specific phenotype, but it would be awkward to characterize it as a genetic change
Line 77- is the issue in DS changes in expression patterns or just expression levels caused by trisomy
Line 137- Is “DNA methylation is commonly thought to explain the multiple phenotypes that present in DS [19].”? Maybe specific phenotypes have been proposed to have a methylation-based mechanism, but I wouldn’t generalize across phenotypes like this. I think gene dosage may be more of an issue than DNA methylation at many chromosome 21 genes
Line 145- used, not use
Line 175- …as well as contribute…not “a contributed” I think there are many more typos and grammatical errors in the manuscript; I will leave this level of editing to the authors and subsequent copy editors.
Line 48 brings up the lack of involvement of the DSCR in facial phenotypes, but later in the manuscript (lines 185-190) there is a discussion of DSCR gene involvement in facial phenotypes. This is very confusing and if it is to be discussed in this review, the contradiction should be addressed.
Line 323- This must be removed: “Click or tap here to enter text.”
Line 385- gene expression is increased
Line 388- this is a run-on sentence that must be split with a period or rephrased
Line 395-399- Are DS patients at increased risk of developing gliomas? This section is confusing.
Line 403- The cited study is misrepresented- Olig gene expression was not inhibited, the gene copy number was returned to disomy by knock out of one endogenous gene. This kind of inaccuracy is not good for a review article and it does a disservice to potential readers.
Line 503-506- This sentence is not clear. Is the point that trisomy 21 is present in leukemias found in people that do not have Down syndrome?
Line 562- Outside of the system designed by the Lawrence lab to use XIST to manipulate a cell culture model of DS, what role does XIST have in causing dysfunction in DS patients? This is very confusing.
I appreciate the effort to try to rewrite the sections on potential therapeutic approaches involving XIST, etc. but it seems strange to propose there approaches and then state that they are not likely to work or be practical. I think they should not be proposed at all.
Comments on the Quality of English LanguageAs I wrote in my critique, there are still a large number of typographical and grammatical errors in this manuscript.
Author Response
I appreciate that the authors have made substantial changes to the manuscript; however, I think there are too many errors remaining to consider publication at this time. I’ve listed a few here:
We thank the author for their comments and have tried to address there concerns in the bullet points below.
Line 55- Is APP a genetic change? It is a gene implicated in a specific phenotype, but it would be awkward to characterize it as a genetic change
We have changed this based on the reviewers comments.
Line 77- is the issue in DS changes in expression patterns or just expression levels caused by trisomy
We thank the reviewer for their comment and have changed this sentence.
Line 137- Is “DNA methylation is commonly thought to explain the multiple phenotypes that present in DS [19].”? Maybe specific phenotypes have been proposed to have a methylation-based mechanism, but I wouldn’t generalize across phenotypes like this. I think gene dosage may be more of an issue than DNA methylation at many chromosome 21 genes
We have toned down this sentence based on the reviewers comments.
Line 145- used, not use
The spelling mistake has been corrected.
Line 175- …as well as contribute…not “a contributed” I think there are many more typos and grammatical errors in the manuscript; I will leave this level of editing to the authors and subsequent copy editors.
We have removed the sentence that does not make sense.
Line 48 brings up the lack of involvement of the DSCR in facial phenotypes, but later in the manuscript (lines 185-190) there is a discussion of DSCR gene involvement in facial phenotypes. This is very confusing and if it is to be discussed in this review, the contradiction should be addressed.
We have addressed this point by removing the reference to the facial phenotype in line 47 and highlighted how the DSCR is not essential for all clinical features.
Line 323- This must be removed: “Click or tap here to enter text.”
This has been removed.
Line 385- gene expression is increased
We have made this change.
Line 388- this is a run-on sentence that must be split with a period or rephrased
A period has been added here.
Line 395-399- Are DS patients at increased risk of developing gliomas? This section is confusing.
We agree with the reviewer that this section is confusing and have hence removed the discussion regarding glioma cell models. DS patient are thought to be at a reduce risk of glial type tumours so this sentence does not help the understanding of the gene expression pattern.
Line 403- The cited study is misrepresented- Olig gene expression was not inhibited, the gene copy number was returned to disomy by knock out of one endogenous gene. This kind of inaccuracy is not good for a review article and it does a disservice to potential readers.
We agree with the reviewer that this statement is not accurate and have corrected the sentence to reflect the mouse model better.
Line 503-506- This sentence is not clear. Is the point that trisomy 21 is present in leukemias found in people that do not have Down syndrome?
We agree with the reviewer that this sentence does not help the development of the understanding of the causes for the clinical phenotype of DS and have removed it.
Line 562- Outside of the system designed by the Lawrence lab to use XIST to manipulate a cell culture model of DS, what role does XIST have in causing dysfunction in DS patients? This is very confusing.
We agree with the reviewer that this sentence is not clear and have removed it.
I appreciate the effort to try to rewrite the sections on potential therapeutic approaches involving XIST, etc. but it seems strange to propose there approaches and then state that they are not likely to work or be practical. I think they should not be proposed at all.
We thank the reviewer for the comment and have removed the suggestion of targeting XIST. We have removed this section from the manuscript.
Reviewer 3 Report
Comments and Suggestions for Authors
Thank you for your attention to the suggestions made. Your manuscript is improved as a result.
Author Response
Thank you very much for this positive feedback, I will share this with the team.
Round 3
Reviewer 2 Report
Comments and Suggestions for Authors
There are still many grammatical errors. I appreciate that the authors responded to many of my specific comments, but not all of those responses were complete. For example, the discussion about cell culture model approaches being pursued as therapeutic approaches was deleted in response to my objection over how those experiments were portrayed and interpreted, but the remaining text was not sufficiently edited in its absence. In my opinion, this manuscript is still not written well enough to be published.
Comments on the Quality of English LanguageI strongly suggest that this manuscript be more carefully edited if it is to be resubmitted for publication.
Author Response
Thank you for reviewing our work again. We have spent ample time thoroughly editing and changing all of the paper to improve the grammar. I have tried to reword the majority of the paper so that is reads better. We hope that you will now consider it sufficiently edited grammatically to allow for publication.